# Chitosan: A Promising Multifunctional Cosmetic Ingredient for Skin and Hair Care

**Eduardo Guzmán** [1,2,*], **Francisco Ortega** [1,2] **and Ramón G. Rubio** [1]

1  Departamento de Química Física, Facultad de Ciencias Químicas, Universidad Complutense de Madrid, Ciudad Universitaria s/n, 28040 Madrid, Spain
2  Instituto Pluridisciplinar, Universidad Complutense de Madrid, Paseo Juan XXIII 1, 28040 Madrid, Spain
*  Correspondence: eduardogs@quim.ucm.es; Tel.: +34-913-944-107

**Abstract:** The cosmetic industry has an undeniable need to design and develop new ecosustainable products to respond to the demands of consumers and international regulations. This requires substituting some traditional ingredients derived from petrochemical sources with new ones with more ecofriendly profiles. However, this transition towards the use of green ingredients in the cosmetic industry cannot compromise the effectiveness of the obtained products. Emerging ingredients in this new direction of the cosmetic industry are chitosan and its derivatives, which combine many interesting physicochemical and biological properties for the fabrication of cosmetic products. Thus, the use of chitosan opens a promising future path to the design of cosmetic formulations. In particular, chitosan's ability for interacting electrostatically with negatively charged substrates (e.g., skin or damaged hair), resulting in the formation of polymeric films which contribute to the conditioning and moisturizing of cosmetic substrates, makes this polymer an excellent candidate for the design of skin and hair care formulations. This review tries to provide an updated perspective on the potential interest of chitosan and its derivatives as ingredients of cosmetics for skin and hair care.

**Keywords:** adsorption; antimicrobial; biopolymer; conditioning; chitosan; coating; layers; moisturizing; polymers

## 1. Introduction

The transition of the cosmetic industry towards "green cosmetics" imposed by the current international regulations, which have banned the use of many chemicals as ingredients in cosmetic products, and the demand of products with improved ecosustainability has led to extensive research aimed towards finding suitable ingredients for substituting traditional ones derived from petrochemical sources [1–7]. This is of paramount importance when the substitution of common polymers used in cosmetic formulations is considered. In fact, polymers present a broad range of applications in cosmetics, e.g., rheological modifiers, emulsifiers, stimuli-responsive reagents, conditioners, film formers, fixations, foam stabilizers, skin-feel beneficial agents, or antimicrobial agents, which allows considering these types of molecules as the most used ingredients in different families of cosmetic products [8].

Currently, the cosmetic industry uses polymers belonging mainly to four different families: (i) synthetic polymers; (ii) polysaccharide-based polymers; (iii) proteins, and (iv) silicones [9]. These polymers play very different roles in cosmetic formulations. However, the substitution of synthetic polymers and silicones with new "green ingredients" is among the most important challenges for the modern cosmetic industry [10]. This has stimulated the research on the use of polysaccharides to cover most of the roles of the rest of polymer families [11], opening new routes to design and develop ecofriendly and biodegradable formulations containing mainly ingredients from renewable natural sources [12,13].

Polysaccharides are complex carbohydrate polymers containing many hydroxyl groups along their backbone that have been exploited in the development of cosmetic and personal care formulations for centuries [14]. This has been possible due to the multiple biological and physicochemical properties of polysaccharides, including biodegradability, biocompatibility, nontoxicity, renewability, and availability [15]. Furthermore, polysaccharides present a better safety profile than synthetic polymers, reducing the hazards and risks to health and the environment associated with their use [16].

Among polysaccharides, it is possible to find a broad range of polymers with different charges (cationic, nonionic, anionic, or amphoteric) and conformations (mainly loose rigid helices or randomized coils) [17,18], which offers a broad variety of properties to the formulators, pushing the use of polysaccharide to cover different cosmetic necessities, including rheology modifiers, conditioners, healing and suspending agents, moisturizers, hydrators, and emulsifiers [19,20]. Unfortunately, to date, there is no clear understanding of the true role of polysaccharides in cosmetic products. Nevertheless, the use of these long-lifespan, green, and raw materials has been rapidly extended in the cosmetic industry [14]. In particular, the use of chitosan and its derivatives has received much attention due to their low production costs and safety character [21]. Moreover, the environmental aspects have been also considered as a main issue for the exploitation of this abundant polysaccharide in cosmetics [22]. It should be noted that chitosan and chitosan derivatives can be currently produced with a broad range of physicochemical and biological properties for their application in cosmetic, hygiene, and personal care products. In particular, the cosmetic industry exploits different chitosan-based compounds, including chitosan hydrochloride, chitosan acetate, chitosan lactate, carboxymethyl chitosan, quaternized derivatives, oligosaccharides, and also chitin sulfate and carboxymethyl chitin [23–25]. Table 1 summarizes some examples of the potential uses of chitosan and its derivatives in the cosmetic industry.

**Table 1.** Potential uses of chitosan and its derivatives in cosmetic industry. Adapted by permission from Springer-Nature from Morin-Crini et al. [23], Copyright (2019).

| Type of Products | Forms | Applications |
|---|---|---|
| Toiletry | Solution | Functional additives |
| Hygiene | Powder | Moisturizers: maintain skin moisture, tone skin |
| Personal care | Film | Thickening agent |
| Skin care | | Hydrating and film-forming agent |
| Oral care | | Role in surfactant stability; stabilize emulsion |
| Dental care | | Antistatic effect |
| Hair care | | Bacteriostatic |
| Cosmeceuticals | | Encapsulating agent |
| | | Delivery systems |
| | | Products: shampoos, creams, skin creams, creams for acne treatment, lotions, bath lotions, nail polish, fixtures, make-up powder, lacquers, nail lacquers, nail enamel, varnishes, hair sprays, hair colorants, and wave agents |
| | | Cleaning products: cleansing milk, face peel, facial toner, soap, and bath agent |
| | | Hair care: elastic film on hair, increase its softness and mechanical strength, improve suppleness of hair, remove oils and sebum from hairs, reduce static electricity in hair, retain moisture, and maintain hair's style |
| | | Oral care, dental care: toothpaste and chewing gum |

This review aims to present an updated perspective on the use of chitosan as a potential ingredient in cosmetic formulations for skin and hair care. This is important because the introduction of chitosan in different cosmetic and cosmeceutical products is one of the most active research areas of the cosmetic industry, and the understanding of their potential activity in new products requires being comprehensively analyzed. It is important to recall that chitosan is a naturally derived polymer, and, hence, before dealing with the potential applications of chitosan in the cosmetic industry, it is necessary to discuss some

of the most fundamental physicochemical and biological aspects of chitosan. It is worth mentioning that the introduction of chitosan in cosmetics as a substitute for traditionally used synthetic polymers provides important benefits to the cosmetic industry. In particular, the natural origin of chitosan promotes the fabrication of most sustainable and safe products. Moreover, the use of chitosan can also contribute to reducing the pollution of the fabrication processes [26].

## 2. Chitosan: Physicochemical and Functional Properties

### 2.1. Physicochemical Properties

The study of chitosan cannot be understood without introducing some of the main characteristics of its most common source: chitin. Chitin is a naturally occurring polysaccharide that can be found in a broad range of living organisms. In fact, chitin is the second most abundant polysaccharide in the world after cellulose, and is the main component of the exoskeleton of many invertebrates, e.g., crustaceans, mollusks, and insects, but it also plays a vital structural role in the cell wall of different fungi and yeasts [27,28]. Chemically, chitin is a crystalline polymer composed of N-acetyl-2-amino-2-deoxy-β-D-glucose units polymerized through (1,4)-links, presenting many structural analogies with cellulose. In fact, the main chemical difference between chitin and cellulose appears at the C2 position, where the hydroxyl groups of cellulose are replaced by acetamide groups in chitin. Moreover, chitin also presents some 2-amino-2-deoxy-β-D-glucose residues (see Figure 1a for molecular structure of chitin). Chitin is a white, hard, inelastic, and nitrogenous polysaccharide, which presents a limited chemical reactivity and has an analogous role to collagens in animals and cellulose in plants [28].

**Figure 1.** Chemical structures of chitin (**a**) and chitosan (**b**) Reprinted from Casadidio et al. [22], with permission under Open access CC BY 4.0 license, https://creativecommons.org/licenses/by/4.0/ (accessed on 22 July 2022).

The physicochemical and biological properties of chitin and chitosan are extremely dependent on the raw material and the methodology used for their isolation. In fact, chitin is insoluble in aqueous solutions and most common organic solvents. However, it presents good solubility in other solvents, including hexafluoroacetone sesquihydrate, hexafluoroisopropanol, and chloroalcohols. Therefore, the utilization of bare chitin as cosmetic ingredients is very limited due to their low solubility in aqueous medium, which makes its handling very difficult [21]. Chitin presents much more applications when it is transformed to chitosan following a partial deacetylation process under alkaline conditions [29]. In fact, the protonation of the amino groups at the C2 position of chitin leads to its deacetylation that allows obtaining chitosan, a cationic polysaccharide with reasonably good solubility in acid aqueous solutions (lactic, acetic, glutamic, and hydrochloric acid solutions) with a pH up to 6.5, which is the $pK_a$ value of chitosan. Unfortunately, when the pH of the solutions is close to physiological conditions, chitosan becomes insoluble and undergoes precipitation as a result of amine deprotonation. Moreover, the addition of inert electrolytes to chitosan solutions or the increase in the polymer molecular weight can also worsen the polymer solubility [30].

On the contrary to that which happens when chitin is considered, chitosan is mainly formed by 2-amino-2-deoxy-β-D-glucose units (see Figure 1b for the molecular structure of chitosan) and a certain number of N-acetyl-2-amino-2-deoxy-β-D-glucose residues. The amount of these residues within the polymer chains is accounted for by the deacetylation

degree (DD), which is defined as the percentage of deacetylated monomers in relation to the total number of monomers in the polymer chain. The DD controls the solubility of the polymer in acid aqueous solutions. In fact, the increase in chitin DD above a threshold value of 50% makes the polymer soluble, changing its structure to that corresponding to cationic chitosan [30]. The evaluation of the DD of chitosan is a very important issue, and different methodologies have been developed for its determination, including different spectroscopies (ultraviolet, infrared, Raman, and proton and carbon nuclear magnetic resonance), gel permeation chromatography, circular dichroism, residual salicylaldehyde analysis, titration methods, elemental analysis, high-performance liquid chromatography, thermal analysis, or mass spectrometry [31,32].

Moreover, the molecular weight can impact the potential applications of polysaccharide polymers. The molecular weight of chitin is generally in the range 400–2500 kDa and depends on the source and the methodology used for its isolation [33,34]. On the other side, the molecular weight of chitosan is lower than that of chitin due to the N-deacetylation process, appearing commonly in the range of 100–500 kDa, and depends on the degree of polymerization (DP). This parameter becomes very important because it influences the chitosan's solubility. In fact, when the $DP \leq 8$, chitosan is soluble in water independently of the deacetylation degree [35].

### *2.2. Functional Properties*

Chitosan presents different functional properties that derive from their biological activity, e.g., anticholesterolemic, wound-healing, anticancer, fungistatic, hemostatic, analgesic, antiacid, antiulcer, or immunoadjuvant [36,37]. In the particular case of the cosmetic industry, chitosan has been commonly used as an excipient and bioactive ingredient. This is possible by exploiting several chitosan properties, including limited toxicity, biocompatibility, and biodegradability [22].

### 2.2.1. Antimicrobial Activity

Chitosan presents good antimicrobial activity against different microorganisms, including bacteria, fungi, and yeast. However, the mechanism underlying chitosan antimicrobial activity is far from clear and there are several hypotheses available trying to account for this important role of chitosan. The first hypothesis suggests that the antimicrobial activity of chitosan is associated with the ionic cross-linking occurring between the polycation and the negatively charged cell surface. This leads to the formation of a very dense layer which hinders the intake of nutrients into the microorganisms, causing their death. A second hypothesis tries to explain the antimicrobial activity of chitosan on the basis of its chelating properties and their influence on organism growth. The last hypothesis considers that low-molecular-weight chitosan can penetrate through cell walls, which facilitates its interactions with deoxyribonucleic acid (DNA), altering some important biological pathways, including ribonucleic acid (RNA) and protein synthesis. It should be noted that the increase in the chitosan charge, i.e., the increase in the DD, favors the antimicrobial role of chitosan, improving its permeation and its electrostatic binding to the membrane [38,39]. Moreover, the increase in the polymer molecular weight and the solution pH is also favorable for the antimicrobial activity of chitosan [40]. In general, the antimicrobial effectiveness of chitosan is higher against Gram-negative bacteria than against Gram-positive ones [41].

### 2.2.2. Antioxidant Activity

Chitosan presents scavenging ability against different radical species including oxygen, e.g., alkyl, superoxide, hydroxyl, and DPPH (2,2-diphenyl-1-picrylhydrazyl). However, as occurred for the antimicrobial activity of chitosan, to date, there is no clear description of the mechanism driving the antioxidant activity of chitosan, which is commonly ascribed to the ability of the hydroxyl and amino groups of chitosan to chelate free metal ions, leading to the formation of stable species [22].

It should be stressed that the antioxidant properties of chitosan are governed by the specific molecular properties of the chosen polymers. In fact, the increase in the DD and the reduction in the molecular weight contribute to the enhancement of the antioxidant properties of chitosan [42]. The role of the latter parameter can be understood considering that the tendency of short chains to form intramolecular hydrogen bonds between their hydroxyl groups is very limited, which results in an increase in the number of hydrophilic groups available for radical scavenging [43].

### 2.2.3. Mucoadhesive Properties

The specific properties of chitosan, in particular its positive charge, favor its interaction with the negative residues of the mucin. This interaction is enhanced for polysaccharides of high DD and high molecular weight [44–46]. Moreover, the derivatization processes of chitosan can also contribute to enhancing the mucoadhesive properties of the obtained polymers in comparison to bare chitosan [44].

### 2.2.4. Penetration Enhancement

Chitosan can act as a penetration enhancer through a modification of the transepithelial electric resistance occurring as a result of the opening and destruction of the epithelial tight junctions. This is possible through the electrostatic binding of chitosan to the cell membrane, which modifies the association of the proteins involved in the tight junctions [47,48]. For instance, Contri et al. [49] pointed out that the skin penetration of acrylic capsules embedded into a chitosan gel can be enhanced in relation to the situation in which chitosan is not in the medium. Thus, chitosan leads to a deeper skin penetration of the cosmetic formulations mediated by the opening of the tight junctions of the stratum granulosum upon its interaction with the positive charges of the polymer. The penetration of chitosan into hair fibers was explored by Kojima et al. [50] by using time-of-flight secondary ion mass spectrometry (TOF-SIMS), evidencing that the positive charge of chitosan controls its penetration into hair fibers. This penetration depends strongly on the degradation degree of hair. Thus, the formation of a cysteic acid group upon bleaching favors the deposition of chitosan in comparison to the situation found for virgin hair.

## 3. Chitosan in Skin and Hair Care

Chitosan has been isolated for a long time, mainly from the skeletons of different crustaceans, including crab, shrimp, and lobster [51]. However, this chitosan is not suitable for cosmetic applications due to the possible disease transmission (zoonosis), and different ethical issues and concerns related to biodiversity and endangered-species protection [52,53]. This has stimulated the cosmetic application of chitosan derived from plants or biotechnological processes. In particular, the isolation of chitosan through the fermentation of fungal cell walls has been developed as a very interesting and economical approach for obtaining this polymer with cosmetically acceptable properties [51].

The current cosmetic uses of chitosan are mainly related to the production of mascaras, hair conditioners, hair foams, and body creams [23]. However, its many functional properties have expanded the research aimed towards the introduction of chitosan in cosmetic and personal care formulations for skin, oral, nail, and hair care [21]. This section will be focused on the discussion of the main aspects related to the use of chitosan for the design of suitable formulations for skin and hair care. These take advantage of the specific properties of chitosan, including its cationicity which favors the interaction with damaged hair fibers and skin; bacteriostatic, fungistatic, and antistatic character; film-forming ability (essential in the hair-conditioning process); moisture retaining (chitosan can retain moisture even in low-humidity environments); and ability for the controlled release of bioactive agents. Moreover, chitosan presents good compatibility with common ingredients of cosmetic formulations, e.g., starch, glucose, saccharose, polyols, oils, fats, waxes, acids, nonionic emulsifiers, and nonionic water-soluble gums [23]. This has led, recently, to the development and commercialization of different cosmetic-grade ingredients based on chitosan

b [54]. Table 2 summarizes some of the currently available commercial cosmetic-grade ingredients based on chitosan.

**Table 2.** Some examples of cosmetic-grade ingredients based on chitosan.

| Commercial Name | Manufacturer | Application |
| --- | --- | --- |
| Hydamer™ | Chitinor AS (Tromsø, Norway) | Film-forming and fixative agent, deodorizing |
| Ritachitosan® | Rita Corporation (Crystal Lake, IL, USA) | Film-forming agent |
| Curasan™ | Chemisches Laboratorium Dr. Kurt Richter GmbH (Berlin, Germany) | Film-forming agent |
| Zenvivo™ | Clariant (Muttenz, Switzerland) | Film-forming agent, antimicrobial, deodorizing, moisturizer |
| KIOsmetine® | Kitozyme (Herstal, Belgium) | Film-forming agent, moisturizer |
| Chitosonic® Acid | Personal Care Products Council (Washington DC, USA) | Antimicrobial, moisturizer |
| ChitoClear™ | Primex Manufacturing Inc. (Langley, BC, Canada) | Film-forming agent |
| Everquat™ Q50H | Sino Lion (Florham Park, NJ, USA) | Shining agent, antidandruff agent, hair growth promoter, anti-hair-loss agent |
| Vinkocos p-6N | Vink Chemicals GmbH & Co. KG (Kakenstorf, Germany) | Film-forming and wetting agent, thickener, stabilizer |
| Jeen-Chitosan | Jeen International (Fairfield, NJ, USA) | Film-forming agent, moisturizer |
| Triozan | Ovensa Inc. (Aurora, ON, Canada) | Penetration enhancer |

Moreover, there are several brands including chitosan in their cosmetic products for different applications [55]. Table 3 summarizes some currently commercialized cosmetic products containing chitosan and their specific applications.

**Table 3.** Some examples of currently commercialized cosmetic products containing chitosan and their specific applications. Adapted from reference [55].

| Product | Manufacturer | Application |
| --- | --- | --- |
| Scalp Purifying Micellar Shampoo | Kristin Ess Hair (Los Angeles, CA, USA) | Shampoo |
| Brazilian Joia Strengthening + Smoothing Shampoo | Sol de Janeiro, Inc. (New York, NY, USA) | Shampoo |
| Extra Gentle Conditioner | Kristin Ess Hair (Los Angeles, CA, USA) | Hair conditioner |
| Herbal Essences Set Me Up Gel | Procter and Gamble (Cincinnati, OH, USA) | Hair-styling gel |
| Re Vamp Mid Length Repair Cream | Vernom Francois (Los Angeles, CA, USA) | Hair serum |
| Anti-Aging Moisture Lotion | Murad LLC (El Segundo, CA, USA) | Skin Care |
| Ultimate Miracle Worker Multi-Rejuvenating Cream | Philosophy (New York, NY, USA) | Skin Care |
| St. Yves Replenishing Mineral Therapy Body Lotion | Unilever (London, UK) | Skin moisturizer |

### 3.1. Chitosan as Ingredient in Skin Care Products

The good antioxidant, cleansing, protecting, humectant, and antioxidant functions of chitosan make it a suitable ingredient for skin care applications. In particular, chitosan has been broadly exploited as an antiaging and moisturizing agent, in ultraviolet protection, in skin cleansing, and as a boosting factor of different essential functions of the skin (protection, absorption, thermal regulation, defense, reservation, and synthesis) [22]. One of the most important aspects of chitosan in relation to its use in skin care products is related to its minimal penetration through the skin, which limits its action mechanism in most of the formulations to the skin/external environment interface [23,56].

Chitosan deposition on the skin surface can contribute to wound tissue healing by creating a network structure and stimulate the synthesis of collagen, maintaining good air permeation. Moreover, it presents good biocompatibility and biodegradability; antibacterial, hemostatic, and anti-inflammatory properties; good absorption of exudate; and promotion of tissue regeneration and skin collagen fiber growth. This favors its uses as an ingredient of a broad range of skin care products [57]. Figure 2 shows a scheme highlighting some of the most common uses of chitosan in skin care cosmetics.

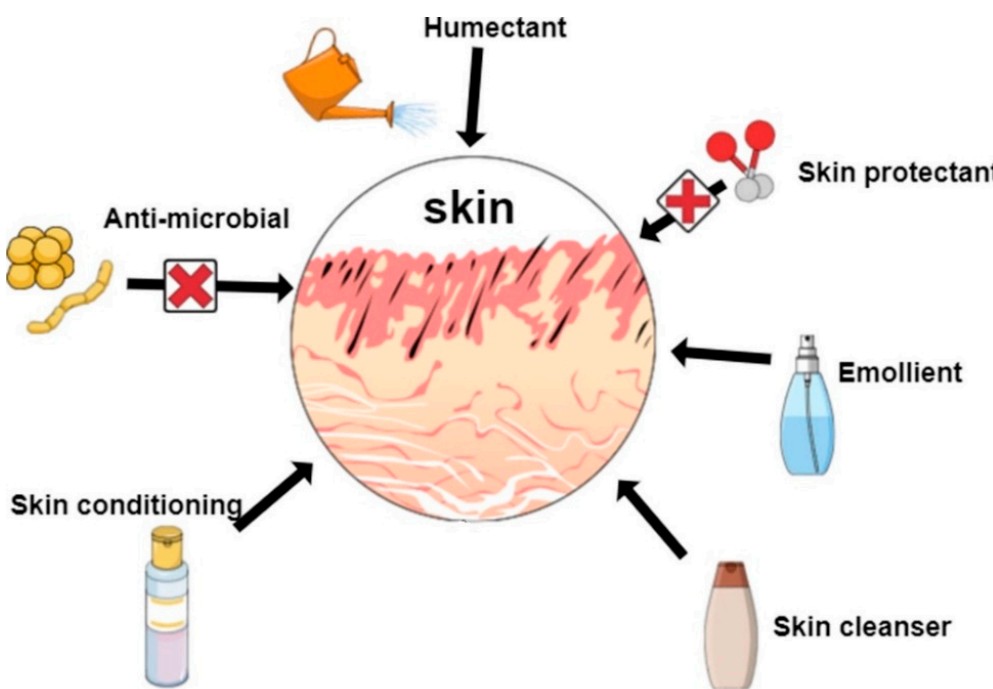

**Figure 2.** Potential uses of chitosan in skin care cosmetics. Adapted by permission of Springer-Nature from Rejinold et al. [56], Copyright (2021).

### 3.1.1. Chitosan Applications as a Humectant and Moisturizing Agent

Humectants are cosmetic formulations or ingredients that contribute to the increase in the water content on the top layers of the skin, whereas moisturizers increase the skin water content, contributing to improving skin softness and smoothness [21]. Therefore, it is possible to consider humectants as a moisturizer ingredient used to replace the natural moisturizing factors existing in the keratin layer of the skin [58].

The role of chitosan as a humectant exploits its cationic character which allows its adsorption on the negatively charged skin surface. In fact, chitosan can adsorb on the negatively charged skin surface, improving the stratum corneum water content and increasing the fluidity of the cell membrane [59]. Moreover, the higher the molecular weight, the higher the moisture retention capacity of the chitosan. This can be understood by considering the higher number of available monomers within the polymer chain, which favors the formation of intermolecular hydrogen bonds. These are responsible for the modulation of the moisture adsorption/retention by chitosan [60].

Chaiwong et al. [61] showed that carboxymethyl chitosan may act as a suitable moisturizing ingredient, and that moisturizing properties are enhanced with the increase in the polymer molecular weight. This moisturizing effect can be explained considering the ability of the polymer to form a hydrated chitosan layer on the skin surface, avoiding water evaporation. On the other side, the moisturizing effect of chitosan also increases with the deacetylation degree [59]. The use of carboxymethyl chitosans as suitable moisturizing ingredients in cosmetics was further studied by Chaiwong et al. [61], who found that this type of polymer may be applied successfully in deodorant creams for obtaining a satisfactory water content in the skin upon application.

The chemical modification of chitosan by adding anionic moieties provides a suitable strategy for enhancing the ability of chitosan for moisture adsorption, reaching a moisturizing effect even better than that provided by hyaluronic-acid-based products. Therefore, anionic modified chitosans are emerging as promising ingredients for ensuring a good adsorption and retention of moisture upon the application of cosmetic products [62].

### 3.1.2. Skin Aging

Skin aging is a very important problem associated with skin degradation as a result of the combined effect of intrinsic aging occurring with time and different external factors, including smoke, UV radiation, and pollution. This induces dryness, relaxation, roughness, and laxity in the skin, which can originate skin hyperpigmentation and the appearance of wrinkles, especially upon extensive exposure to UV radiation (photoaging) [63]. Chitosan has been proven as a very promising ingredient to reduce the problem associated with skin aging. In fact, the ability of chitosan, especially high-molecular-weight polymers, to form films upon its application on the skin surface minimizes the cutaneous water loss and enhances the skin's mechanical properties (elasticity and smoothness), playing a very important role as a moisturizing agent [64]. Kong et al. [65] demonstrated that the biological activity of chitosan can be exploited for minimizing the undesirable changes associated with extensive exposure to UV radiation. For instance, the application of chitosan can contribute to reducing the macroscopic and histopathological damage occurring in skin through the stimulation of the collagen production pathways. Moreover, chitosan can also inhibit the production of postinflammatory cytokines and improve the activity of different antioxidant enzymes and the moisture level of the skin. Therefore, chitosan can be exploited to prevent UV-induced skin dryness, epidermal hyperplasia, and wrinkle formation. This is possible because chitosan can increase the activity of certain enzymes with an antioxidant role and suppress proinflammatory cytokine production, which in turn inhibits the degradation of collagen fibers.

Libio et al. [66] pointed out that the ability of chitosan solubilized in citrate buffer to form films on the skin surface can be exploited as a very interesting strategy for stratum corneum exfoliation. This takes advantage of chitosan's bioadhesive character for reducing the cohesion of cells, and their subsequent detachment, which can contribute to preventing skin aging by inducing cell proliferation and the regeneration of the corneum layer. Moreover, chitosan also presents an important role in increasing collagen density, reducing lines and wrinkles. The main advantage of this strategy in comparison to wound-healing processes is associated with the ability of chitosan to stimulate the interactions between proteins and cells. This induces cell proliferation, the formation of a permeable barrier (re-epithelialization), and angiogenesis.

Chen et al. [67] designed composites of quaternized carboxymethyl chitosan and organic montmorillonite as a novel antiaging ingredient for cosmetic creams. This ingredient favors the adsorption and retention of moisture thanks to its layered structure and high number of hydrophilic groups, providing very good protection against UV radiation.

### 3.1.3. UV Protection

Repeated exposure to UV radiation can induce oxidative stress and inflammatory disequilibrium, leading to skin photoaging. This is characterized by the formation of wrinkles, skin dryness, irregular pigmentation, and laxity. Moreover, the role of exposure to UV radiation in the emergence of skin cancer cannot be neglected [21]. Therefore, skin protection against the undesirable effects associated with UV exposure is essential for ensuring the functional integrity of living organisms [56].

The UV-visible spectrum of chitosan presents absorption bands below 400 nm. This makes possible the use of chitosan as a suitable ingredient for sunscreen cosmetics because the UV rays associated with solar radiation presents wavelengths in the range 320–400 nm and 290–320 nm in the case of UV-A and UV-B, respectively [21].

Chitosan's power as sunscreen was proved by Verma et al. [68] who treated dyed cotton fabric with chitosan, and found that the ultraviolet protection factor increased more than 20% in relation to the value corresponding to the bare dyed cotton fabric. Moreover, treatment with chitosan extended the durability of the photoprotection. It should be noted that the UV-protection associated with the use of chitosan is strongly dependent on the specific characteristic of the used polymer, including source, molecular weight, and

deacetylation degree [22]. Unfortunately, to date, there are no systematic studies dealing with how the characteristics of chitosan can influence its performance as sunscreen.

Morsy et al. [69] fabricated sunscreen gels consisting of hydroxyapatite nanoparticles homogeneously dispersed within a chitosan matrix. The deposition of a film of this hybrid system on the skin surface was found to be a suitable strategy for minimizing the effects of exposure to UV radiation.

### 3.1.4. Skin Cleansing

Skin-cleansing procedures aim to remove from the skin any substance that can be deposited upon their exposure to ambient air or the application of cosmetic products. The use of chitosan and some of its derivatives as skin cleansers is possible by exploiting their cationic nature for designing carriers of active ingredients contributing to the cleansing process. Thus, the interaction between the positive charges of the chitosan backbone and the anionic charges of the skin surface can be exploited as a very promising tool for ensuring the targeted release of cleaners [70]. Moreover, Theerawattanawit et al. [71] demonstrated that chitosan gels may be exploited for reducing sebum levels without any side effects.

The ability of chitosan in the form of nanoparticles to control sebum levels was studied by Tangkijngamvong et al. [72] who found a significant decrease in the sebum levels after one week of the treatment with the cosmetic formulation containing chitosan particles. Moreover, skin oiliness underwent a continuous decrease even after four weeks of the formulation application. Sebum removal associated with chitosan is associated with the ability of chitosan to form complexes with sebum, contributing to its removal. Simultaneously, chitosan can form a film that prevents the deposition of sebum on the skin surface.

### 3.1.5. Antibacterial Role

The most accepted mechanism explaining the antibacterial activity of chitosan upon application on the skin can be explained considering that positively charged chitosan can interact with the negatively charged bacterial cell wall, which leads to its weakening as a result of a shrinkage process. This results in the inactivation of the bacterial cell, which is strongly dependent on the molecular weight and charge density of the chitosan chains. In fact, the higher the chitosan molecular weight, the stronger the antimicrobial power [56]. This antibacterial role was proved by Verma et al. [68] who found a reduction in the bacterial proliferation upon chitosan treatment higher than 90%.

Chi et al. [73] showed that the use of formulations containing chitosan microneedles for skin treatment reduces the inflammatory response associated with bacterial proliferation, accelerating collagen deposition, angiogenesis, and granulation tissue formation. Moreover, the formation of a chitosan hydrogel-like layer provides a significant protective effect, acting as a barrier to prevent future bacterial infections.

Burkatovskaya et al. [74] showed that chitosan acetate presents higher antimicrobial effectiveness against *Pseudomonas aeruginosa* and *Proteus mirabilis* than alginate and silver sulfadiazine. This high activity against Gram-negative bacteria is explained by their fast action mechanism mediated by the destabilization of the lipopolysaccharide bacterial cell membrane and its subsequent permeabilization, leading to the leakage of cellular content and avoiding a further proliferation of bacteria.

### 3.2. *Chitosan as an Ingredient in Hair Care Products*

Chitosan is not only a very good ingredient for skin care products, providing also many benefits to the hair, improving hair hydration, helping to rebuild damaged hair, and providing a healthy shine. This has stimulated the use of chitosan in a broad range of hair care products, including shampoos, rinses, permanent wave agents, hair colorants, styling lotions, hair sprays, and hair tonics [75,76]. This intensive use of chitosan is mainly associated with chitosan's ability to improve the rheological properties of cosmetic formulations or enhance the adhesion of specific components to the hair [21].

One of the most common hair care aspects in which cationic polymers, such as chitosan, are involved is the conditioning process. In fact, the proteinic structure of damaged hair is characterized by a denaturalized structure with negative charges which cannot be repaired by biological processes, and, hence, it is necessary to use physicochemical methods for its temporal reparation. This is possible by taking advantage of the ability of cationic polymers to form a film on the surface of negatively charged surfaces [3,77–79]. In particular, the interaction of chitosan and its cationic derivatives with the negatively charged keratin surface of damaged hair fibers leads to the formation of transparent elastic films, which increase hair softness and strength, minimizing the hair damage induced by mechanical, thermal, or environmental stresses [80].

Hernández-Rivas et al. [5] explored the ability of chitosan to form conditioning films on the surface of negatively charged surfaces mimicking the hair fibers, and found that even though chitosan presents a good ability to deposit on negatively charged surfaces, its performance as a conditioner is, in general, worse than that found for traditional conditioning polymers such as poly(diallyl-dimethyl-ammonium chloride), polyquaternium-6, and JR400, polyquaternium-10. However, a correct choice of the molecular weight and the deacetylation degree of the used chitosan is of paramount importance in improving its conditioning effects, and allows exploiting the full potential of chitosan in hair cosmetics.

Sionkowska et al. [81] contributed to the understanding of the role of chitosan in conditioning formulations. They explored the effect of ternary blends including chitosan, collagen, and hyaluronic acid, which appear as very promising formulations for hair-conditioning purposes, and found that chitosan can contribute to improving the mechanical properties, surface free energy, and stability in aqueous conditions on the conditioning deposits, contributing to an improvement in the appearance of hair fibers. Moreover, the combination of collagen, hyaluronic acid, and chitosan as a conditioning formulation increases the elasticity and resistance of hair fibers against damage. In fact, the use of the ternary mixture increases the Young's modulus and the maximum elongation before the breaking of fibers [82].

## 4. Chitosan as Delivery Systems

Chitosan-based delivery systems have gained interest in the cosmetic industry due to their capacity to release active ingredients at specific targets in a controlled way [28]. In particular, the use of chitosan nanoparticles emerges as a very promising alternative because this type of material can interact with the skin lipid layers [83], which has stimulated their use for the topical and transdermal delivery of different active compounds [84].

Panonnummal et al. [83] demonstrated that the use of formulations based on chitosan nanogel particles loaded with an antipsoriatic active ingredient (clobetasol) favors the transdermal flux of the drug in comparison to a conventional formulation, ensuring a higher retention of the active ingredient within the deep layers of the skin. This provides evidence of the potential benefits of nanogel for the delivery of active ingredients in skin care applications. Ta et al. [58] designed chitosan gel particles which present high moisture adsorption ability and low moisture retention capacity. This allows their use for the encapsulation of hydrophilic cosmetic ingredients, and the subsequent release of the encapsulated molecules within a hydrophobic environment such as the skin. Moreover, this type of particle presents good capacities for penetrating through the skin and gathering in the dermis layer, offering suitable properties for the dermal delivery of active ingredients.

Abd-Allah et al. [85] fabricated chitosan particles loaded with nicotinamide, and tested their use against acne. These particles exhibited very strong adhesion to the skin, providing a high deposition of nicotinamide in the different layers of the skin, which led to an important reduction in the inflammatory acne lesions.

Chitosan particles have been exploited in hair care applications. In particular, Matos et al. [86] produced chitosan particles loaded with minoxidil to face alopecia problems (topical treatment). These particles presented high accumulation in the hair follicles and

contributed to a good drug release profile, ensuring a relevant therapeutic concentration for more than 12 h.

Chitosan particles loaded with eugenol and carvacrol have shown good properties for ensuring the preservation of cosmetic formulations against microbial contamination [87]. Similar preservation of cosmetic formulations can be obtained by loading other preservatives such as thymoquinone [88]. Therefore, the encapsulation ability and release profile of chitosan particles present are of paramount importance in the design of natural preservatives of cosmetic products.

## 5. Toxicity Aspects of Chitosan

Chitosan is commonly considered as a nontoxic, biocompatible polymer [89], and has been approved for regulatory agents for different uses. However, chitosan cannot be included as a generally recognized as safe (GRAS) material. This is because the specific characteristics of the used chitosan as well as its chemical modifications can influence the safety profile of the specific polymer. Therefore, the safety of the use of chitosan in cosmetics should be assessed case by case, even though in most cases, chitosan can be considered safe for cosmetic applications [90]. The inclusion of chitosan in cosmetic products may modify the biodistribution profiles of the different compounds in the formulation. For instance, the interactions with cells may be altered due to the presence of charged particles in chitosan. Therefore, the number of positive charges contained in the chitosan molecules, which depends on the molecular weight and deacetylation degree of the polymer and the pH of the medium, may modify chitosan interactions with cells and the microenvironment, and, hence, the control of the above parameter is essential for modulating the potential toxicity of formulations containing chitosan. Moreover, the nature of the products and the application mode can also influence the potential toxicity of chitosan in cosmetics [90].

## 6. Prospects and Challenges of Chitosan as Cosmetic Ingredients

The above discussion has pointed out that chitosan is an emerging ingredient for fulfilling some of the main requirements of the "green cosmetic" industry. This is reflected in the rapid growth in the chitosan market for cosmetic applications. In fact, in recent years, it has been possible to find many papers and patents dealing with the use of chitosan in the cosmetic industry. Unfortunately, there is a limited number of commercialized products containing chitosan, and when it is used as a cosmetic ingredient, in most cases, its bioactivity and physicochemical properties are not completely exploited. This is in part because it is extremely difficult to access chitosan with enough purity and reliability of its sources. Moreover, the natural origin of chitosan makes it very difficult to obtain, at the industrial level, raw materials with the same characteristics, and when it is possible, the production costs become very elevated (higher than that of petroleum-derived polymers), which is a very important drawback to the introduction of chitosan as an ingredient in the cosmetic industry [23]. Therefore, it is mandatory to seek ecological and economical sustainable alternatives for producing chitosan with suitable characteristics for use in the cosmetic industry. This can pass through the design of synthetic routes providing chitosan, or its derivatives, with enough purity and properties to be considered as a suitable ingredient for cosmetic products. Moreover, it is essential to obtain extensive knowledge of the structural and physicochemical aspects of chitosan, which is not always easy due to the above-mentioned heterogeneity of isolated chitosan [91]. In fact, one of the most important questions that emerges in front of cosmetic formulators is related to the suitability of a specific chitosan sample for a fixed application. Therefore, a detailed characterization of the samples before their application in cosmetic products is needed [21].

In addition to the aspects related to the quality of the chitosan as a cosmetically acceptable ingredient, it is mandatory to perform in vivo tests, which can provide evidence of the effectiveness of chitosan as an active ingredient for cosmetics, giving also information about the possible harmful effect of this ingredient [21,23].

## 7. Concluding Remarks

The undeniable need for cosmetic products with a more ecosustainable profile and higher safety has pushed the cosmetic industry towards a progressive substitution of many ingredients derived from petrochemical sources. This is only possible by introducing natural-derived polymers in cosmetics, which may contribute to reducing the environmental impact of the cosmetic industry and the negative effects on human health of specific ingredients. The use of natural-derived polymers in cosmetics exploits their physicochemical and biological properties for a broad range of applications, including skin and hair care and make-up, but also, they can be exploited as stabilizers and modifiers, resulting in the fabrication of highly marketable products which can fulfil consumer demands and the current international regulations.

Among natural polymers, chitosan and its derivates are receiving particular and increasing attention, contributing to the change in the traditional paradigm of the cosmetic industry. This has been possible because chitosan presents suitable properties to be used in several types of cosmetics, including skin and hair care products, nail lacquers and lotions, or moisturizers for lips and skins. Moreover, chitosan presents suitable properties for providing sunscreen with water resistance. On the other side, the antimicrobial properties of chitosan allow its use in the fabrication of different deodorants or products for acne treatments. This multifunctionality of chitosan in cosmetic applications is strongly related to the specific physicochemical and biological properties of this polymer and its derivates.

Despite the broad range of potential uses of chitosan in the cosmetic industry, there are several unclear issues that require additional research for ensuring the optimal utilization of this biopolymer and its derivates. The main challenge is associated with the optimization and scaling of the isolation and purification processes of chitosan from its sources. Moreover, the development of suitable strategies for the characterization of the raw material from a physicochemical and biological perspective will be essential for chitosan exploitation. On the other side, one of the main limitations towards the use of chitosan in cosmetics derives from the absence of in vitro and in vivo tests of the new formulations, which limits the evaluation of their efficacy. In this review, we have attempted to provide an updated perspective of the current trends in the use of chitosan for skin and hair care cosmetics. It is expected that this can be useful for cosmetic formulators to optimize the use of chitosan and its derivatives in new cosmetic products that can be marketed and fulfill the demands of the cosmetic green challenge.

**Funding:** This work was funded by MICIN under grant PID2019-106557GB-C21, and by the E.U. on the framework of the European Innovative Training Network-Marie Sklodowska-Curie Action NanoPaInt (grant agreement 955612).

**Conflicts of Interest:** The authors declare no conflict of interest. The funders had no role in the design of the study; in the collection, analyses, or interpretation of data; in the writing of the manuscript; or in the decision to publish the results.

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
