# Peer review of "Chitosan: A Promising Multifunctional Cosmetic Ingredient for Skin and Hair Care"

_cosmetics, doi:10.3390/cosmetics9050099_

Round 1

Reviewer 1 Report

I recommend that the authors will make a Table In section 3 in order to summarize different applications of chitosan. For example, for UV protection, what is the screening efficiency %? For antibacterial roles, what is killing efficiency %? What is the physical form of chitison-related sample (gel, solution, etc)? How they were prepared? These will make this work to be more straightforward to the readers and thereby help to get more citations.

Author Response

I recommend that the authors will make a Table In section 3 in order to summarize different applications of chitosan. For example, for UV protection, what is the screening efficiency %? For antibacterial roles, what is killing efficiency %? What is the physical form of chitison-related sample (gel, solution, etc)? How they were prepared? These will make this work to be more straightforward to the readers and thereby help to get more citations.

The reviewer suggestion is very interesting. Unfortunately, the heterogeneity of chitosan, referred to structure and properties, makes very difficult to summarize the mentioned information within a Table.

We thank to the reviewer for the comments, they were very useful for improving the manuscript.

Reviewer 2 Report

Chitosan: a promising multi-functional cosmetic ingredient for 2 skin and hair care by Eduardo Guzmán et al., provide an update perspective to the potential interest of  chitosan, and its derivates, as ingredients of cosmetics products for skin and hair care, is a well written review articles. 

I have the following recommendation: 

1. Abstract is scientifically not clear. Authors are advised to revise the abstract section and be clear to the point. 

2.  Introduction section need more literature review with latest literature. 

3. Kindly add one or two sentence... how this polymer is better than other synthetic or semisynthetic polymer  used in cosmetics..

4. Is there any cosmetic grade Chitosan available or only pharmaceutical grade are used in cosmetics? 

5. Is there any marketed product based on chitosan available, kindly add one table along with brands name and manufacturer name.

6. Kindly add one section dealing with future prospects and challenges for chitosan as a cosmetics ingredients.

After these minor modification, manuscript can be accepted. 

Author Response

Chitosan: a promising multi-functional cosmetic ingredient for 2 skin and hair care by Eduardo Guzmán et al., provide an update perspective to the potential interest of  chitosan, and its derivates, as ingredients of cosmetics products for skin and hair care, is a well written review articles. 

I have the following recommendation: 

  1. Abstract is scientifically not clear. Authors are advised to revise the abstract section and be clear to the point. 

We have modified the abstract for clarity.

  1. Introduction section need more literature review with latest literature. 

The modification of the introduction will make the rest of the review very repetitive, and thus we prefer to maintain the introduction in the current form.

  1. Kindly add one or two sentence... how this polymer is better than other synthetic or semisynthetic polymer  used in cosmetics..

Following the reviewer recommendation, we have included a comment related to the advantage of chitosan.

  1. Is there any cosmetic grade Chitosan available or only pharmaceutical grade are used in cosmetics? 

We apologize with the reviewer; this information was included in the previous version of the manuscript, but it was not clear enough. We have modified the manuscript to improve the clarity. They are summarized in Table 2.

  1. Is there any marketed product based on chitosan available, kindly add one table along with brands name and manufacturer name.

Following the reviewer recommendation, we have added information of commercial products containing chitosan.

  1. Kindly add one section dealing with future prospects and challenges for chitosan as a cosmetics ingredients.

We have included a section including the information suggested by the reviewer.

After these minor modification, manuscript can be accepted. 

We thank to the reviewer for the comments, they were very useful for improving the manuscript.

Reviewer 3 Report

It is suggested to add the following sections.

Cost effecitveness in comparision to similar polymers.

Toxicity aspects of the polymer.

Further studies needed for establishing it as a suitable polymer for cosmetics.

Author Response

It is suggested to add the following sections.

Cost effecitveness in comparision to similar polymers.

Unfortunately, it is not possible to include a detailed discussion about the commented aspect because up to date there are no a true evaluation of the costs of chitosan in comparison to other polymers in cosmetic applications.

Toxicity aspects of the polymer.

Following the reviewer suggestion, we have included a brief section discussing some issues above the toxicity of chitosans.

Further studies needed for establishing it as a suitable polymer for cosmetics.

This has been included within the new section 6.

We thank to the reviewer for the comments, they were very useful for improving the manuscript.